# LLMs Leak Training Data Beyond Verbatim Memorization via Membership Decoding

## Abstract

Extracting training data from large language models (LLMs) exposes serious memorization issues and privacy risks. Existing attacks extract data by generations, followed by membership inference. However, extraction attacks do not guide such generations, and the extraction scope of member data is limited to the greedy decoding scheme. Only verbatim memorized member data is being audited in this process. And a majority of member data remains unexplored, even if it is partially memorized. In this work, we define a new notion of memorization, $k$-amendment-completable, to measure the degree of partial memorization. Greedy decoding can only extract $0$-amendment-completable sequences, which are verbatim memorized. To address the limitation in generation, we propose a membership decoding scheme, which introduces membership information to guide the generation process. We formulate the training data extraction problem as an iterative member token inference problem. The token distribution is calibrated with membership information at each generation step to explore member data. Extensive experiments show that membership decoding can extract novel member data that haven't been studied before. The proposed attack manifests that the privacy risk in LLMs is underestimated.

## 1 Introduction

Large Language Models (LLMs) have shown remarkable capabilities in text generation Guo et al. (2025); Comanici et al. (2025). However, their ability to memorize and to reproduce training data raises significant concerns about extracting private and sensitive information. Works Carlini et al. (2021); Yu et al. (2023); Hayes et al. (2025); Biderman et al. (2023b) have studied the memorization of LLMs and estimated the corresponding privacy risk under data extraction attacks. Verbatim memorization is a well-studied notion, where the model can complete a training prefix with the exact same training suffix by greedy decoding. Attackers can easily extract and identify these memorized sequences by prompting a training prefix, followed by membership inference attacks (MIAs) Carlini et al. (2021; 2022b).

Even though greedy decoding is simple and efficient, it limits the generation scope to verbatim memorized sequences only in extraction attacks. Due to this decoding limitation, partially memorized sequences are unexplored but remain experiencing significant extraction risks. The generation diversity can be improved by introducing bias and randomness in the decoding strategies, such as beam search decoding Wu et al. (2016) top-K sampling Fan et al. (2018), and temperature sampling Radford et al. (2019). Recent works Hayes et al. (2025); Yu et al. (2023) have studied the memorization amplification brought by randomness, generating multiple candidate suffixes on a given training prefix. Multiple generations and sequence-level MIAs increase the chance of the member hit, but the computational cost as well. Most importantly, these MIAs after generation methods succeeded only when the member is extracted in the generation, which is not optimal. The membership information of the prefix is not fully utilized to guide the generation. The member data that can be extracted with some membership guidance during decoding remains unexplored.

In this work, we explore the possibility of extracting member data directly by introducing membership bias. The key research question is:

> *Can LLMs generate partially memorized training data by themselves with a membership-guided decoding strategy?*

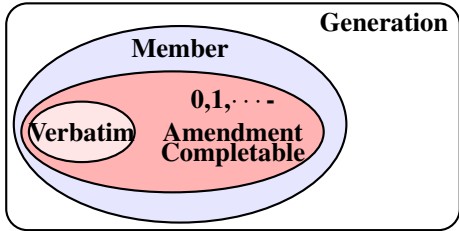

Figure 1: Relationship between Generation, Member, $k$-amendment-completable, and Verbatim memorization.

To answer this question, we define a new notion of memorization, $k$-amendment-completable, to measure how much a suffix is partially memorized by LLMs given its training prefix. Specifically, it computes the number of tokens $k$ that need membership guidance to change during generation. Verbatim memorized sequences are $0$-amendment-completable sequences that can be extracted without any membership guidance. Our goal is to extract more partially memorized sequences, i.e., $k$-amendment-completable sequences with $k > 0$, as shown in Figure 1.

By translating the training data extraction problem as a training data completion the problem, we decompose problem into iterative token-level membership inference attacks during generation. We identify member data in-generation rather than post-generation. Following this motivation, we propose **Membership Decoding** framework, a new decoding strategy that guides the next token prediction with membership information. It allows us to leverage MIA scores to calibrate the prediction distribution at each generation step. Accordingly, we propose a novel token-level membership inference attack based on maximizing a posterior probability of observing the member prefix.

In this work, we give an affirmative answer. Greedy decoding only reveals a small fraction of extractable memorization. Studies based on it underestimate the training data extraction risk. Membership decoding can generate member data beyond fully memorized sequences, extracting $k$-amendment-completable sequences with $k > 0$. It allows us to perform a privacy study on the unexplored member data. Our contributions are threefold:

1. We define a new notion of memorization in LLM, $k$-amendment-complement, measuring the partially memorized sequence that can be generated by LLMs with $k$ token amendment.

2. We propose a membership decoding scheme that formulates the training data extraction problem as iterative membership inference attacks, allowing us to leverage membership information to generate member data.

3. We define a novel token-level membership attack to generate member data, unifying existing MIA methods. The extraction of partially memorized sequences manifests the underestimated extraction risk in existing literature.

## 2 RELATED WORK

### 2.1 MEMBERSHIP INFERENCE ATTACK

Membership inference attacks (MIAs) aim to decide whether a specific data point was included in the training dataset of a target model Shokri et al. (2017); Yeom et al. (2018); Carlini et al. (2022a); Ye et al. (2022); Zarifzadeh et al. (2023). MIAs are first introduced for auditing machine learning algorithms Shokri et al. (2017). This method and subsequent works are based on the training of many reference models to calibrate the MIA score for the target model behavior.

MIAs on LLMs have been considered a challenging problem Duan et al. (2024). Methods can be categorized into training-based MIA and training-free MIA. Training-based MIAs, like LiRA Carlini et al. (2022a); Rossi et al. (2025), train many IN models and OUT models to calibrate the MIA scores for a specific target sample, which is expensive for LLM. Training-free MIAs Shi et al. (2023); Mattern et al. (2023); Mireshghallah et al. (2022); Xie et al. (2024) analyze LLM signals themselves. *Min-k* Shi et al. (2023) and its variant *Min-k++* Zhang et al. (2024a) calibrate the $k$-lowest token likelihood, which is insensitive to token changes out of the minimum $k$. *DC-PDD*

Zhang et al. (2024b) calibrate token probability with token frequency distribution. Rather than identifying true members, they define non-member detectors by catching outlier signals. *Neighbor-comparison* Mattern et al. (2023) catches the overfitting signal by referring to neighbor data, which is inefficient involving massive generations from reference model. *RECALL* Xie et al. (2024) computes the likelihood shift amount with a non-member prefix. These MIAs catch the average token signal for sequence MIAs, and thus are not useful when the target extraction is failed in the default generation. In this work, we design membership-guided generation for better extraction.

## 2.2 TRAINING DATA EXTRACTION

Data extraction attacks aim to recreate a training data point from a target model. It reveals severe privacy risks of leaking sensitive and personally identifiable information (PII). Unintentional memorization is recognized as a main source of extraction. Existing extraction attacks can be categorized as training-based and training-free attacks.

Training-free attacks perform membership inference attacks after massive generations. Carlini et al. (2021) performed loss-based MIA on text sequences generated with greedy decoding. The extractable sequences are limited to verbatim memorized sequences. Tiwari & Suh (2024); Hayes et al. (2025) expanded the generation scope by probabilistic sampling, taking the top-k sampling method to explore members. Yu et al. (2023) adjusted the probability distribution of tokens with repetition penalty and temperature to allow diversity in massive generation. However, performing membership inference attacks on massive generations is expensive and inefficient for extraction.

Training-based attacks train assistant models to extract training data from target models Ozdayi et al. (2023); Wang et al. (2024). Ozdayi et al. (2023) proposed to learn a model adapter in the form of a soft prompt to extract training data. Wang et al. (2024) proposed to learn a soft prompt generator to dynamically enhance the extraction capability. However, these training-based attacks require white-box access to the target model for gradient computation. They also need a large amount of training data to train the adapter or generator. They are not applicable in black-box settings where only output logits or probability distributions are available.

Apart from the extraction of exact suffix, approximate extractions relax the constraints to allow partial matching Biderman et al. (2023a), n-gram matching Ippolito et al. (2022), and approximate string matching Kassem et al. (2024). In this work, we focus on the exact suffix extraction, which is more challenging and more useful in real-world applications.

## 3 THREAT MODEL

### 3.1 MEMORIZATION

We define a member sequence $x$ if there exists an exact same sequence $x$ in the training dataset of a model $f$. Due to the next token prediction training objective in LLMs, any member's prefix $x_{<t}$, the leading $t-1$ tokens of $x$, is a member as well.

**Definition 3.1** ($k$-extractible). A suffix $s$ is $k$-extractible if it is generated by the model $f$ when prompted with a prefix $p$ of length $k$, and $[p\|s]$ is in the training set of $f$.

$k$-extractible is the first memorization notion to estimate the privacy risk of a sequence Carlini et al. (2021). It studies verbatim memorization by varying the length of prefix $k$, with which LLMs can reproduce member data by greedy decoding. The greedy decoding strategy chooses the token with the highest likelihood as a continuation during generation. In most prior works Carlini et al. (2021; 2022b), greedy decoding is the default setting for **one-shot extraction**. Sampling introduces randomness for multiple suffix generations, known as **multi-shot extraction**. A member sequence is $(n, p)$-extractible Hayes et al. (2025) if it is generated in $n$ trials with probability $p$, measuring memorization under probabilistic decoding. However, multiple generations and sequence-level MIAs are expensive and inefficient for extraction. More importantly, these notions only consider the number of tokens needed in the prefix to extract the suffix, ignoring the possibility of partial memorization on a given prefix.

| $k$-amendment-completable | LLMs(Prefix) Generation | k-extractable |
|:---:|:---:|:---:|
| Target Sequence $\rightarrow$ | Alice's ID is 123 456 789 | $k = \|$ Alice's ID is 123 456 7$\|$ |
| $k = 0$ | Alice's ID is 123 456 7**89** | $\Rightarrow$ **Risky prefix** |
| $k = 1$ | Alice's ID is 123 456 **666** | $\Rightarrow$ Safe prefix |
| $k = 1$ | Alice's ID is 1**23 456 666** | $\Rightarrow$ Safe prefix |
| $k = 2$ | Alice's ID is **000 000 000** | $\Rightarrow$ Safe prefix |

Table 1: Examples for memorization notions. Red indicates an error decoding token and Green indicates an amendment token. $k$-amendment-completable can capture the risk accurately for different given prefix, while k-extractable underestimate the risk of a short prefix.

**Definition 3.2** ($k$-amendment-completable). A suffix $s$ is $k$-amendment-completable if model $f$ needs $k$ amendments to generate $s$ during greedy decoding when prompted with the prefix $p$, and $[p, s]$ is in the training set of $f$.

To capture the extraction risk of a member sequence in one-shot extraction, we introduce a new memorization notion, $k$-amendment-completable. Prefixes with smaller $k$ remain highly memorized and are more susceptible to extraction attacks. $k$-extractable only considers the last incorrect token, while $k$-amendment-completable considers the total number of incorrect tokens in suffix of a given prefix. For example, in Table 1, given a prefix *"Alice's ID is 1"*, LLMs generate a non-member suffix *"23 456 666"*. However, the member is exposed by replacing non-member token "6" with the member token "7" during generation, showing a high privacy risk of this partial memorization. This phenomenon is also observed in Hayes et al. (2025). In experiments, as shown in Figure 2b, we find that $k$ decreases in average as model size increases, showing heavier memorization in larger models. And the extraction success rate decreases as $k$ grows. It manifests that $k$-amendment-completable notion is valid and more practical in estimating real-world extraction risk and memorization.

### 3.2 PROBLEM DEFINITION

We define the data extraction attack in language models as a training data completion problem:

**Definition 3.3** (Training Data Extraction). Given a target language model $f$ trained for next token prediction on dataset $D$ and any prefix $p$ from a member sequence $x = [p, s] \in D$, the goal is to design a mechanism $g$ to generate the target suffix $s$:

$$g(p, f) = s.$$

Carlini et al. (2021) defined their extraction mechanism as greedy decoding $g := argmax_i(f(p)_i)$. Hayes et al. (2025) introduced sampling into the generation mechanism $g := Sampling_i(f(p)_i)$. However, none of them takes membership information during generation. We explore the member information of the prefix to calibrate the next token prediction distribution during decoding. **Our goal is to extract member data that is $k$-amendment-completable with $k > 0$, broadening generation scope to partial memorization.**

### 3.3 THREAT MODEL

Following the literature, we consider the threat model defined in Carlini et al. (2021).

**Victim Definition** The victim provides black-box access to the target language model and returns the logits or probability at every token prediction on the query sequence.

**Adversary's Capabilities** Adversary can query the probability of the next token $v_i \in V$ on any sequence $x_{<t}$. The weight and intermediate prediction of LLM are hidden. An adversary can sample the member prefix $p = x_{<t}$ with an unknown suffix $s$.

**Adversary Objective** The goal of the adversary is to extract the member suffix given the member prefix. A stronger attack can extract more $k$-amendment-completable sequences with larger $k > 0$. The extraction fills the gap between training data and training data that is decoded by greedy search.

# 4 METHOD

In this section, we explain our membership decoding formulation to address the training data extraction problem. We also propose a new score for context-aware token-level membership inference, distinguishing a member token from a set of non-member tokens in the vocabulary.

The first challenge is that the candidate space of possible member suffixes is huge. It grows exponentially in the suffix length $N - n$: $|V|^{N-n}$, where $N$ is the length of the target sequence and $n$ is the prefix length. The second challenge is that the membership inference attack score is not accurate when the data is not verbatim memorized. The score is computed as the likelihood of suffix given prefix $P_f(x_{n,\cdots,x_{N-1}}|x_{<n}) = \prod_{t=n}^{N-1} P_f(x_t|x_{<t})$.

## 4.1 MEMBERSHIP DECODING

We address the first challenge by formulating the training data extraction problem as a training data completion problem. We decompose it into a sequence of membership inference attacks during the generation. The key insight is that the prefix of any length of a member sequence is also a member. We can perform MIAs on the next member token prediction given the member prefix. In particular, due to the auto-regressive nature of LLMs, the training objective of a training sequence is a series of token predictions with cross-entropy loss $l$:

$$L(x_{<N}) = \sum_{t=1}^{N-1} l(x_t|f(x_{<t})).$$

Each prefix $x_{<t}$ serves as a member as the target sequence $x_{<N}$ in the training dataset. The model is trained to predict the next member token $x_t$ given its prefix $x_{<t}$. Thus, we can perform token-level membership inference attacks during generation to complete the member suffix.

We alter the usual token generation to member token generation for the extraction of the member sequence auto-regressively. At each step, a member sequence is identified by a membership inference attack among the candidates. The candidate space is reduced to the size of vocabulary $|V|$ at each step and grows linearly with the suffix length. The Membership Decoding process is defined as follows: Given a prefix $x_{<n}$, and a target sequence length $N - 1$,

1. Construct the candidate member set at step $t \geq n$: $\{c_i^t = [x_{<t}, v_i]\}_{i=1}^{|V|}$.

2. Compute the membership score $Score_{MIA}(c_i^t, f)$ for each candidate given its prefix $x_{<t}$ is a member of $f$.

3. Select the candidate with a maximum score $g(x_{<t}, f) := \arg\max_i(Score_{MIA}(c_i^t, f))$.

4. Iterate over steps 1-3 until $t = N - 1$.

We define the mechanism $g$ as a next member token prediction function that takes the prefix $x_{<t}$ and the model $f$ to predict the next member token $x_t$ until the member sequence $x_{<N}$ is completed.

$$g(x_{<t}, f) = x_t, \quad \forall t = n, n+1, \cdots, N-1$$

In each next token prediction, the mechanism $g$ leverages membership score to select a member token as the next token. This process calibrates the default token distribution to favor member tokens, bringing minor overhead to the default decoding process. Thus, we name this framework as Membership Decoding, aiming to pop up the member token during the generation. It allows us to explore different membership attacks by varying the membership score in Step 2.

In the next section, we explain our design for membership score. Greedy decoding is a special case of membership decoding. It implements Loss-based MIA Yeom et al. (2018), computing membership score as the likelihood of a candidate token given the prefix:

$$Score_{Loss}(f, c^t) = Pr_f(x_t|x_{<t}).$$

However, this MIA score fails to calibrate the hardness of the target sample Carlini et al. (2022a). Without calibration, the membership scores are not accurate when the data is partially memorized.

## 4.2 MAXIMIZE A POSTERIOR AS MIA SCORE

In this section, we propose our score computation method in performing membership inference attacks. Given $x_{<t}, \forall t < n$ are members of the model $f$, we aim to evaluate the probability of observing $(x_{<t}, x_t = v_i)$ as a member sequence. To address the second challenge, we need to calibrate the membership score to better distinguish the member sequence from the non-member sequences. The key insight is that if $v_m$ is the member continuation of $x_{<t}$, the posterior probability of observing the member prefix $x_{<t}$ should be higher than given a non-member continuation $v_{nm}$.

We define MIA score as the probability of "$f$ is trained on $x_{<t}$" given "$f$ is trained on $(x_{<t}, x_t = v_i)$": $P(f, x_{<t}|x_t = v_i)$. However, we cannot directly compute this probability of a candidate sequence $c_i^t$ given the prefix $x_{<t}$. It requires computing the probability of observing "$f$ is trained on $x_{<t}$" over all possible training datasets. Thus, we apply Bayes' rule to compute the posterior probability of a candidate sequence $c_i^t$ given the prefix $x_{<t}$:

$$v_m = \arg\max_{v_i} P(f, x_{<t}|x_t = v_i), \tag{1}$$

$$(\text{with Bayes}) \quad = \arg\max_{v_i} \frac{P(f, x_{<t})P(x_t = v_i|f, x_{<t})}{P(x_t = v_i)}, \tag{2}$$

$$(\text{Independent}) \quad \propto \arg\max_{v_i} \frac{P(x_t = v_i|f, x_{<t})}{P(x_t = v_i)}, \tag{3}$$

where prior $P(f, x_{<t})$, the probability of "$f$ is trained on $x_{<t}$", is independent to candidates and left out. We approximate likelihood with next token prediction probability on $f$:

$$P(x_t = v_i|f, x_{<t}) = Pr_f(x_t = v_i|x_{<t}). \tag{4}$$

The evidence $P(x_t = v_i)$ is the total probability of "observing the token $v_i$ as the next token given the prefix $x_{<t}$". It is computed by marginalizing all possible models $h$ evaluated on the prefix $x_{<t}$:

$$P(x_t = v_i) = \sum_{h, x_{<t}} P(h, x_{<t})P(x_t = v_i|h, x_{<t}). \tag{5}$$

We can resort to open-weight LLMs as the reference models to compute the evidence. However, it is challenging to find reference LLMs $h$ that are trained on the same target sequence $x_{<N}$. And thus the estimation is biased to the models that are not trained on sequence $x_{<N}$, i.e., OUT models. We calibrate this bias by assuming that the estimation on IN models is an affine function of the estimation in OUT models as RMIA Zarifzadeh et al. (2023):

$$P(x_t = v_i|h_{IN}, x_{<t}) = a \cdot P(x_t = v_i|h_{OUT}, x_{<t}) + (1 - a) \quad \in [0, 1], \tag{6}$$

where $a$ is a hyper-parameter to control the calibration strength. $a = 0$ overestimates the likelihood of IN models with 1, assuming the model perfectly fits the target sequence. While $a = 1$ underestimates the likelihood of IN models, assuming no probability difference after training with the target sequence. Finally, the evidence is approximated as follows:

$$P(x_t = v_i) \approx \frac{1}{2}((1 + a) \cdot P(x_t = v_i|h, x_{<t}) + (1 - a)). \tag{7}$$

We take $a = 0.5$ throughout the experiment as a trade-off. The additional overhead is the generation process on the reference model $h$. However, the reference model is usually much smaller than the target model, and the generation is efficient. The membership decoding process is efficient with minor overhead compared to the default decoding process. For a robust calibration, we take the top-20 tokens from the target model as the candidate set at each step in experiments. It avoids the computation instability when the token probability is too tiny. And a token with a tiny probability is unlikely to be a member token. Overall, we compute our token-level membership score as follows:

$$Score_{MIA}(c_i^t, f) = \frac{2 \cdot Pr_f(x_t = v_i|x_{<t})}{(1 + a) \cdot Pr_h(x_t = v_i|x_{<t}) + (1 - a)} \tag{8}$$

In summary, we define the membership score by calibrating the next token prediction probability with the evidence of observing token $v_i$ as the member continuation over all possible models $h$. Greedy decoding Carlini et al. (2021) takes no calibration as a Loss-based MIA method. Reference-based Mireshghallah et al. (2022) methods calibrates with the likelihood of a reference model, which is $a = 1$ in our case. DC-PDD Zhang et al. (2024b) calibrates with $P(x_t = v_i)$ by taking it as a prior distribution and computing it as the token frequency distribution. However, it is not accurate and requires access to the training dataset. We unify these MIA methods in our membership decoding framework by varying the membership score.

## 5 EXPERIMENT

In this section, we first verify our memorization notion $k$-amendment-completable in the existing LLM models. Then we evaluate the effectiveness of membership decoding in generating partially memorized training data.

### 5.1 EXPERIMENT SETUP

We evaluate on the MIMIR benchmark Duan et al. (2024). The MIMIR dataset is created from the Pile datasetGao et al. (2020) for MIA evaluation. We take 7 datasets from different domain: "Pile CC", "ArXiv", "PubMed Central", "HackerNews", "DM Mathematics", "GitHub", "Wikipedia", each of which contains $1,000$ member sequences. We also take the "Full Pile" dataset containing $10,000$ training examples. The last 10 tokens are taken as the target suffix and the rest tokens as the prefix.

For all the experiment, we take the transformer models introduced in Pythia Biderman et al. (2023b). The target models are Pythia-1B, 1.4B, 2.8B, 6.9B, 12B, and the reference model is the smallest model, Pythia-70m, consistent with Shi et al. (2023); Xie et al. (2024). Smaller models tend to memorize less of the training data Biderman et al. (2023a). For Pythia-12B, the prefix length is limited to 300 tokens due to the memory limitation. We leave the result of OPT family Zhang et al. (2022) and OLMo family Groeneveld et al. (2024) in the Appendix. All experiments are conducted on two NVIDIA RTX-TITAN GPUs with 24GB of memory. The model weights and datasets are loaded from HuggingFace Wolf et al. (2020). No randomness is introduced in both generation and membership inference attacks. And all results are reproducible.

We compare four decoding baselines:

- **Loss** (greedy Yeom et al. (2018)): It selects the token with the highest probability at each step, defining MIA score as Eqn. 4.
- **Ref** ($a = 1$ Mireshghallah et al. (2022)): It calibrates MIA score with reference OUT model by likelihood ratio, i.e., $a = 1$ in Eqn. 8, catching the token that has the greatest increase rate compared to the reference model.
- **Minus**: It computes the MIA score as the probability difference, catching the token that has the greatest probability gains compared to the reference model.
- **Our** ($a = 0.5$): It compares the token distribution with the total probability calibrated by an affine transformation as Eqn. 8.

The token-insensitive MIA methods like Min-k% Shi et al. (2023) and prefix-insensitive MIA methods like DC-PDD Zhang et al. (2024b) are not applicable in our membership decoding framework, as they cannot distinguish member tokens from non-member tokens at each step. The performance is measured by the sequence-level exact match accuracy in one-shot extraction.

To evaluate the next member token prediction task, we construct two settings: **Hard** and **Easy**. The **Hard** setting requires performing the next 10 member tokens extraction. The **Easy** setting requires performing the next single member tokens prediction at the failure case of greedy decoding. Evaluation data is constructed by the failure case of greedy decoding, where the token with the highest probability is not a member token.

### 5.2 MEMORIZATION WITH $k$-AMENDMENT-COMPLETABLE

In this section, we verify the validity of our memorization notion $k$-amendment-completable on existing LLMs, and estimate the privacy risk of extraction for each model and dataset. To evaluate $k$ for each sequence, we query the target model with the prefix and continue the generation by amending the incorrect token with the suffix token. The $k$ is the number of amended tokens in the greedy decoding process. The target sequence is extracted when the attacks perform the same operation. Computing this notion of memorization is efficient as the generation process.

**GitHub dataset presents a serious memorization than other domains.** To evaluate the extraction risk across data domains, we present the $k$ distribution among different domains on Pythia-6.9B in Figure 2a. The results on the other model scales are similar. The HackerNews dataset is mainly

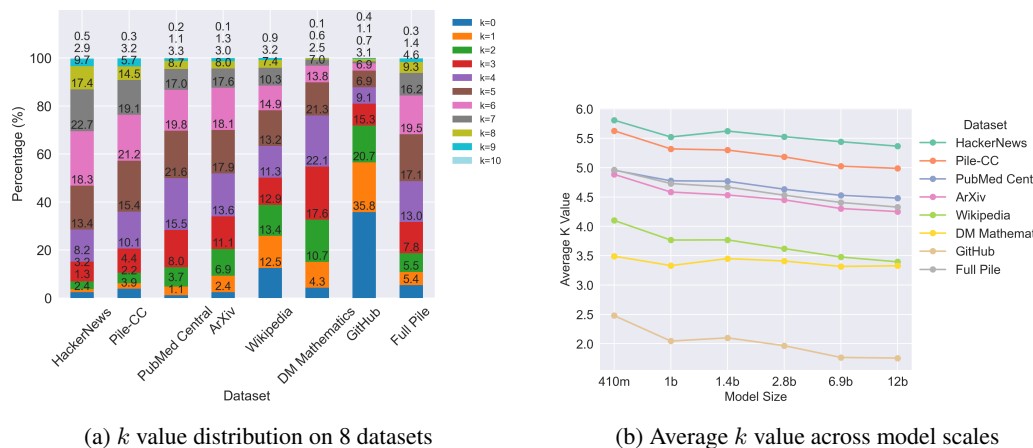

(a) $k$ value distribution on 8 datasets

(b) Average $k$ value across model scales

Figure 2: Memorization analysis using $k$-amendment-completable notion. (a) $k$ distribution shows serious memorization of GitHub than other domains. (b) The average $k$ value decreases as model size increases, indicating that larger models memorize more training data.

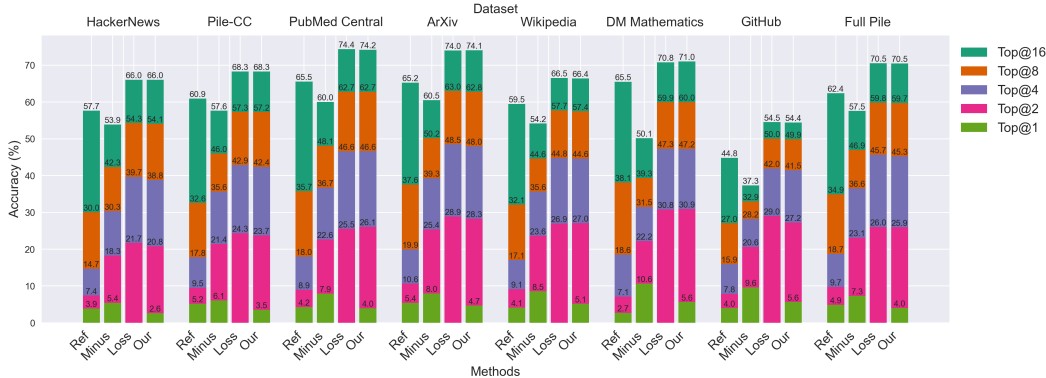

Figure 3: Easy: Result of single token MIA on each $k$-amendment-completable sequence on Pythia-6.9B. The membership information can recover the member token when the greedy decoding fails.

composed by 6 and 7-amendment-completable sequences, while the GitHub dataset has a majority of 0 and 1-amendment-completable sequences. Our notion indicates that the Pythia model memorizes more GitHub samples than HackerNews samples. And GitHub could be more susceptible to privacy attacks. Our observation is supported by their MIA vulnerability in Duan et al. (2024). It suggests that the $k$-amendment-completable concept is a fine-grained measure of memorization. Sequences with small $k$ are highly memorized and face a higher extraction risk.

**Models at larger scale memorize more training data.** To evaluate the relationship between memorization and model scale, we also evaluate the average $k$ value over datasets for each model scale in Figure 2b. We introduce a smaller model with 410m parameters for a full-scale comparison. The average $k$ value decreases when the model scales up across all the datasets. This phenomenon suggests that larger models memorize more training data than smaller models. It supports us to take smaller models as reference models in membership decoding, as they memorize less training data.

## 5.3 RESULTS IN EASY SETTING

In **Easy** setting, We answer the research question **how well the membership decoding score can recover the member token when the greedy decoding fails**. It evaluates the case where the attacker has prior knowledge on the candidate token set but greedy decoding returns an outlier, like those in constrained decoding Melcer et al. (2024). We measure the top@m hits ($m = 1, 2, 4, 8, 16$) of the membership inference attack at a single token. For every sequence in this setting, greedy

Table 2: One-shot Extraction Accuracy (%) by $k$, Method, Dataset, and Model Size

| $k$ | Method | HackerNews | | | | | Pile-CC | | | | | PubMed Central | | | | | ArXiv | | | | |
|---|---|---|---|---|---|---|---|---|---|---|---|---|---|---|---|---|---|---|---|---|---|
| | | 1b | 1.4b | 2.8b | 6.9b | 12b | 1b | 1.4b | 2.8b | 6.9b | 12b | 1b | 1.4b | 2.8b | 6.9b | 12b | 1b | 1.4b | 2.8b | 6.9b | 12b |
| 0 | Ref | 0.0 | 0.0 | 0.0 | 0.0 | 0.0 | 0.0 | 0.0 | 0.0 | 0.0 | 0.0 | 0.0 | 0.0 | 0.0 | 0.0 | 0.0 | 0.0 | 0.0 | 0.0 | 0.0 | 0.0 |
| | Minus | 16.0 | 22.2 | 11.5 | 29.2 | 29.2 | 28.1 | 26.7 | 27.3 | 48.7 | 52.4 | 0.0 | 12.5 | 0.0 | 9.1 | 15.4 | 7.1 | 12.5 | 10.5 | 16.7 | 35.3 |
| | Our | 96.0 | 92.6 | 80.8 | 95.8 | 91.7 | 93.8 | 93.3 | 87.9 | 92.3 | 90.5 | 69.2 | 100.0 | 77.8 | 81.8 | 76.9 | 71.4 | 68.8 | 68.4 | 58.3 | 76.5 |
| 1 | Ref | 0.0 | 0.0 | 0.0 | 0.0 | 0.0 | 0.0 | 0.0 | 0.0 | 0.0 | 0.0 | 0.0 | 0.0 | 0.0 | 0.0 | 0.0 | 0.0 | 0.0 | 0.0 | 0.0 | 0.0 |
| | Minus | 6.2 | 0.0 | 10.0 | 7.7 | 6.2 | 0.0 | 0.0 | 5.3 | 0.0 | 9.1 | 0.0 | 0.0 | 0.0 | 0.0 | 0.0 | 1.8 | 0.0 | 3.6 | 4.3 | 5.4 |
| | Our | 6.2 | 0.0 | 10.0 | 7.7 | 6.2 | 0.0 | 0.0 | 0.0 | 4.5 | 9.1 | 0.0 | 2.9 | 0.0 | 5.4 | 5.4 | 7.3 | 5.3 | 3.6 | 5.8 | 8.1 |
| 2 | Ref | 0.0 | 0.0 | 0.0 | 0.0 | 0.0 | 0.0 | 0.0 | 0.0 | 0.0 | 0.0 | 0.0 | 0.0 | 0.0 | 0.0 | 0.0 | 0.0 | 0.0 | 0.0 | 0.0 | 0.0 |
| | Minus | 0.0 | 0.0 | 0.0 | 0.0 | 0.0 | 0.0 | 0.0 | 0.0 | 0.0 | 0.0 | 0.0 | 1.5 | 1.2 | 0.0 | 1.0 | 0.0 | 0.0 | 0.0 | 0.9 | 0.0 |
| | Our | 0.0 | 0.0 | 0.0 | 0.0 | 0.0 | 2.6 | 0.0 | 0.0 | 0.0 | 0.0 | 0.0 | 0.0 | 0.0 | 0.0 | 0.0 | 0.0 | 0.0 | 1.0 | 0.9 | 0.0 |

| $k$ | Method | Wikipedia | | | | | DM Mathematics | | | | | GitHub | | | | | Full Pile | | | | |
|---|---|---|---|---|---|---|---|---|---|---|---|---|---|---|---|---|---|---|---|---|---|
| | | 1b | 1.4b | 2.8b | 6.9b | 12b | 1b | 1.4b | 2.8b | 6.9b | 12b | 1b | 1.4b | 2.8b | 6.9b | 12b | 1b | 1.4b | 2.8b | 6.9b | 12b |
| 0 | Ref | 0.0 | 0.0 | 0.0 | 0.0 | 0.0 | 0.0 | 0.0 | 0.0 | 0.0 | 0.0 | 0.0 | 0.0 | 0.0 | 0.0 | 0.0 | 0.0 | 0.0 | 0.0 | 0.2 | 0.0 |
| | Minus | 15.2 | 19.6 | 15.0 | 19.2 | 22.7 | 2.5 | 2.9 | 2.3 | 7.0 | 9.1 | 32.7 | 27.7 | 32.8 | 42.5 | 43.0 | 38.5 | 30.2 | 32.4 | 39.6 | 38.3 |
| | Our | 88.6 | 85.3 | 80.4 | 80.0 | 77.3 | 72.5 | 71.4 | 74.4 | 79.1 | 79.5 | 93.6 | 89.0 | 90.4 | 91.9 | 88.9 | 90.2 | 86.7 | 86.7 | 88.8 | 88.3 |
| 1 | Ref | 0.0 | 0.0 | 0.0 | 0.0 | 0.0 | 0.0 | 0.0 | 0.0 | 0.0 | 0.0 | 0.0 | 0.0 | 0.0 | 0.0 | 0.0 | 0.0 | 0.0 | 0.0 | 0.0 | 0.0 |
| | Minus | 1.5 | 1.7 | 1.4 | 2.2 | 3.6 | 0.0 | 0.0 | 1.1 | 0.0 | 0.0 | 12.4 | 6.2 | 6.8 | 6.8 | 7.1 | 2.7 | 2.5 | 3.9 | 2.9 | 4.0 |
| | Our | 4.6 | 5.9 | 4.3 | 10.4 | 5.8 | 4.8 | 4.3 | 4.5 | 4.7 | 3.7 | 15.7 | 14.3 | 10.4 | 8.2 | 11.9 | 5.3 | 3.9 | 6.4 | 5.5 | 5.7 |
| 2 | Ref | 0.0 | 0.0 | 0.0 | 0.0 | 0.0 | 0.0 | 0.0 | 0.0 | 0.0 | 0.0 | 0.0 | 0.0 | 0.0 | 0.0 | 0.0 | 0.0 | 0.0 | 0.0 | 0.0 | 0.0 |
| | Minus | 0.0 | 0.0 | 0.0 | 0.0 | 0.0 | 0.0 | 0.0 | 0.0 | 0.0 | 0.0 | 2.4 | 0.0 | 1.4 | 1.3 | 0.0 | 0.6 | 0.6 | 0.1 | 0.6 | 0.2 |
| | Our | 0.9 | 0.0 | 0.0 | 0.0 | 0.0 | 0.0 | 0.0 | 0.5 | 0.0 | 0.6 | 1.6 | 0.7 | 1.4 | 1.3 | 0.0 | 0.3 | 0.4 | 0.3 | 0.5 | 0.2 |

decoding fails for top@1 by definition. Examples in this setting could leak the non-member information as prior knowledge to attackers, as the most probable token is a non-member. We leave the rigorous evaluation in the **Hard** setting.

**MIA scores can identify member tokens when greedy decoding fails.** The experiment results are shown in Figure 3. All three membership scores can rescue some member tokens (Top@1) when the greedy decoding fails. Our score is able to rescue greedy decoding on Top@1 while maintaining a comparable accuracy on Top@16. Compared to Ref without calibration, this result suggests that the total probability calibration in Eq. 7 is effective to keep more member tokens. The Top@16 accuracy of Loss score is around 70%, and our selection of top 20 tokens can cover member token in the majority of failure case. It manifests that extraction attacks could generate a member token with a high probability. Thus, training data could face a high risk of being extracted, even if it is partially memorized.

## 5.4    Result on Hard Setting

In **Hard** setting, we answer the research question **can LLMs generate partially memorized training data with the proposed extraction attacks?** We evaluate the membership decoding on generating the next 10 consecutive member tokens on each group of $k$-amendment-completable sequences. The extraction difficulty increases with a larger $k$, as more non-member tokens scores a lower loss. With a larger $k$, membership decoding is required to amend more tokens during generation for a successful extraction. The Loss baseline is left out as the group $k$ is defined by it. The experiment results are shown in Table 2.

**LLMs can recover partially memorized sequences with membership guidance in decoding.** Our method extract sequences on $k = 1, 2$ with the proposed MIA score. They represent a qualitative breach of privacy that existing methods (greedy decoding) entirely miss. At the same time, our method maintains a high accuracy on the $k = 0$ sequences as the usual greedy decoding. The overall accuracy drops as $k$ increases. The large $k$ sequences are less memorized and under-optimized by the model. The membership signal becomes weaker and the extraction becomes more challenging. Compared to Ref with 0 accuracy, the calibration on total probability in Eq. 7 improves the overall utility of the membership decoding. The GitHub dataset also presents a higher vulnerability to extraction attacks, which is consistent with the observation of average $k$ value in Fig. 2a.

## 5.5 PARAMETER ANALYSIS ON CALIBRATION $a$

We analyse the impact of calibration parameter $a$ in Eqn. 6 on the extraction accuracy on Pythia-2.8b model in Table 3. A smaller $a$ tends to overestimate the likelihood of IN models, while a larger $a$ tends to underestimate it. The accuracy on $k = 0$ sequences increases as $a$ decreases, as the member tokens are favored more during generation. However, the accuracy on $k = 2$ sequences drops as $a$ decreases, as non-member tokens are also favored more. A trade-off is reached at $a = 0.5$, which maintains a high accuracy on both $k = 0$ and $k = 2$ sequences.

Table 3: One-Shot Extraction Accuracy (%) by $k$ and Calibration Parameter $a$

| $k$ | $a$ | HackerNews | Pile-CC | PubMed Central | ArXiv | Wikipedia | DM Mathematics | GitHub |
|---|---|---|---|---|---|---|---|---|
| | | 2.8b | 2.8b | 2.8b | 2.8b | 2.8b | 2.8b | 2.8b |
| 0 | 0 | 84.6 | 97.0 | 77.8 | 78.9 | 89.7 | 84.1 | 95.3 |
| | 0.25 | 80.8 | 90.9 | 77.8 | 78.9 | 86.9 | 81.8 | 93.7 |
| | 0.5 | 80.8 | 87.9 | 77.8 | 68.4 | 80.4 | 74.4.7 | 90.4 |
| | 0.75 | 76.9 | 69.7 | 77.8 | 63.2 | 72.0 | 54.5 | 83.0 |
| | 1 | 0.0 | 0.0 | 0.0 | 0.0 | 0.0 | 0.0 | 0.0 |
| 1 | 0 | 9.1 | 0.0 | 0.0 | 5.2 | 2.9 | 2.4 | 6.0 |
| | 0.25 | 9.1 | 0.0 | 0.0 | 5.2 | 2.9 | 2.4 | 7.9 |
| | 0.5 | 10.0 | 0.0 | 0.0 | 3.6 | 4.3 | 4.5 | 10.4 |
| | 0.75 | 9.1 | 5.6 | 0.0 | 6.9 | 9.4 | 3.6 | 11.6 |
| | 1 | 0.0 | 0.0 | 0.0 | 0.0 | 0.0 | 0.0 | 0.0 |
| 2 | 0 | 0.0 | 0.0 | 0.0 | 0.0 | 0.0 | 0.0 | 0.0 |
| | 0.25 | 0.0 | 0.0 | 0.0 | 0.0 | 0.9 | 0.0 | 0.7 |
| | 0.5 | 0.0 | 0.0 | 0.0 | 1.0 | 0.0 | 0.5 | 1.4 |
| | 0.75 | 0.0 | 0.0 | 1.3 | 2.0 | 0.0 | 0.5 | 1.4 |
| | 1 | 0.0 | 0.0 | 0.0 | 0.0 | 0.0 | 0.0 | 0.0 |

## 5.6 POTENTIAL DEFENSE

To mitigate the extraction risk from membership decoding, obscuring the necessary MIA signal or increasing the required number of amendments (k) such that extraction becomes infeasible. Differential privacy (DP) training **?** can be applied during model training to limit the amount of information about any individual training example that the model can memorize. However, DP may degrade model performance if not carefully tuned. Model editing techniques **?** can be employed to remove specific memorized sequences from the model. By applying model editing techniques to increase the number of amendments of a specific member sequence. For blackbox model with API access, limiting the returned logits (e.g, returning only top@2) or adding noise to the logits can paritially mitigate the attack by corrupting the MIA signal. Future work could explore more effective defense mechanisms against membership decoding attacks.

## 6 CONCLUSION

Membership inference attacks and data extraction attacks have been studied to measure the privacy risk of language models. However, the set of data points that could be extracted successfully using data extraction attack is much smaller than the set of data points whose membership can be successfully inferred. In this paper, we fill this gap by introducing membership decoding. We show that by guiding the data extraction attack using advanced membership inference techniques we can generate training data. Another limitation of existing extraction attacks is that they only focus on generating verbatim memorization. As we show in this paper, our method enables reconstructing member data, even though it doesn't fully memorized by the model in generation. Our results enable a novel direction to study LLMs memorization on partially memorized data whose risk have been underestimated in the literature.

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

# A  APPENDIX

## A.1  OPT AND OLMO RESULTS

Table 4: One-shot Extraction Accuracy (%) by $k$, Method, Dataset on OPT models

| $k$ | Method | HackerNews | | | Pile-CC | | | PubMed Central | | | ArXiv | | |
|---|---|---|---|---|---|---|---|---|---|---|---|---|---|
| | | 1.3b | 2.7b | 6.7b | 1.3b | 2.7b | 6.7b | 1.3b | 2.7b | 6.7b | 1.3b | 2.7b | 6.7b |
| 0 | Ref | 0.00 | 0.00 | 0.00 | 3.10 | 0.00 | 7.70 | 0.00 | 0.00 | 0.00 | 0.00 | 0.00 | 3.70 |
| | Minus | 33.3 | 21.1 | 29.4 | 50.0 | 45.9 | 64.1 | 20.0 | 29.4 | 33.3 | 45.7 | 22.7 | 14.8 |
| | Our | 93.3 | 89.5 | 94.1 | 93.8 | 89.2 | 89.7 | 60.0 | 70.6 | 75.0 | 80.4 | 86.4 | 74.1 |
| 1 | Ref | 0.00 | 0.00 | 0.00 | 0.00 | 0.00 | 0.00 | 0.00 | 0.00 | 0.00 | 0.00 | 0.00 | 0.00 |
| | Minus | 0.00 | 0.00 | 0.00 | 0.00 | 6.70 | 7.70 | 0.00 | 2.30 | 0.00 | 6.40 | 5.60 | 1.60 |
| | Our | 0.00 | 8.30 | 0.00 | 0.00 | 13.3 | 11.5 | 6.20 | 2.30 | 4.00 | 7.40 | 6.50 | 11.5 |

Table 5: One-shot Extraction Accuracy (%) by $k$, Method on OLMo-7b model

| K | method | Dolma |
|---|---|---|
| 0 | Ref | 0.00 |
| | Minus | 8.29 |
| | Our | 89.86 |
| 1 | Ref | 0.00 |
| | Minus | 0.40 |
| | Our | 4.90 |

To better evaluate the effectiveness of membership decoding under different model architectures and datasets, we also conduct experiments on OPT Zhang et al. (2022) and OLMo Groeneveld et al. (2024) families. The reference model is the smallest model in each family, i.e., OPT-125m and OLMo-1.3b. For OPT family, we take three model scales: OPT-1.3b, OPT-2.7b, and OPT-6.7b. We evaluate on four datasets: HackerNews, Pile-CC, PubMed Central, and ArXiv. For OLMo family, we take OLMo-7b as the target model and evaluate on a Dolma dataset example subset Groeneveld et al. (2024). We sample 5000 examples from the Dolma_v1.6 example dataset and limit the prefix length to 90 tokens due to the memory limitation. The other experiment settings are consistent with those in Section 4.1.

The experiment results are shown in Table 4 for OPT and Table 5 for OLMo. Among all three membership scores, our score outperforms others on both model families. It suggests that the total probability calibration in Eq. 7 is effective across different model architectures and datasets. The results also indicate that membership decoding is a general framework for training data extraction attacks. Also, the privacy risk from partially memorized data is prevalent across different model architectures and datasets.

