# OpenReview forum: "LLMs Leak Training Data Beyond Verbatim Memorization via Membership Decoding"
_ICLR.cc/2026/Conference — Submitted to ICLR 2026_

### Official Review · Reviewer_2NzT · 2025-10-27

**Soundness:** 1
**Presentation:** 2
**Contribution:** 3
**Rating:** 2
**Confidence:** 3

**Summary:**

I list the main contributions of this work as following:

- Introduction of a new memorization notion termed "k-amendment-completable," which quantifies partial memorization by measuring how many tokens require amendment during greedy decoding to generate the actual training sequence.
- A membership decoding framework that treats the training data extraction problem as an iterative sequence of token-level membership inference attacks. Rather than performing membership inference after generation, this approach affects the generation process itself using membership information.
- A novel token-level membership inference attack score based on maximizing the posterior probability of observing the member prefix, which calibrates token distributions using reference models and unifies several existing MIA methods under a common framework.

**Strengths:**

- This paper builds on a strong theoretical framework for understanding partial memorization in language models with proper mathematical explanation
- Identifying and quantizing non-verbatim memorization has been an important problem statement in AI security and privacy for long and this paper tackles this exact problem, making this work significant for the field
- Proposes a new way to identify non-verbatim memorization in LLMs, overcoming one of the primary limitations that of attacks which use greedy decoding only

**Weaknesses:**

- There is a lack of ablations in this study and the evaluation is restricted to the Pythia model family, which represents older and smaller-scale architectures compared to contemporary models
- Table 2 is very unclear, why are most of the positions are blank, these missing results seriously puts into question the validity of the results of this study

Overall, this research paper presents a great novel idea and framework but fails to present solid empirical evidence for it's effectiveness. The scope of the experiments is very limited and the results presented do not seem strong enough.

There is a HARD setting mentioned in the paper, however, no results could be found for this setting in the paper

**Questions:**

Kindly let us know if you could present any more results/values to support this paper

Could you provide a reasoning for the somewhat similar performance of this method to Minus on k=1 for HackerNews, Pile-CC

Why weren't other open-source models like OLMo et cetera tried?

---

> ### Author Response · Authors · 2025-11-21
>
> **Summary of Changes**: We have conducted additional experiments on the OLMo model and OPT model family family as requested, clarified the notation in Table 2, and provided the explicit ablation results below.
>
> 1. **Clarification on Table 2 and the "Hard" Setting** We apologize for the confusion regarding Table 2.
>     * The symbol "-" in Table 2 represents a 0.0% accuracy (complete failure) of the baseline methods, not missing data. This highlights that while baselines fail to recover any suffixes in the Hard setting, our method achieves meaningful extraction rates. We will replace all "-" with "0.0" in the final revision to ensure clarity.
>     * Table 2 is the evaluation of the Hard setting (as defined in Section 5.4). We will revise the caption to explicitly state: "Table 2: Extraction accuracy in the Hard setting."
> 2. **Lack of Ablations** We performed an ablation study to isolate the impact of our Total Probability Calibration ("Ref"). Removing the Total Probability Calibration causes a significant performance drop (recover no member on GitHub dataset) validating the necessity of our scoring mechanism. The extraction accuracy (%) results on GitHub are presented below.
>
>     | k   | Method | Pythia- | 1b   | 1.4b | 2.8b | 6.9B | 12B  |
>     | --- | ------ | ------------ | ---- | ---- | ---- | ---- | ---- |
>     | 0   | Ref    |              | 0    | 0    | 0    | 0    | 0    |
>     |     | Our    |              | 93.6 | 89.0 | 90.4 | 91.9 | 88.9 |
>     | 1   | Ref    |              | 0    | 0    | 0    | 0    | 0    |
>     |     | Our    |              | 15.7 | 14.3 | 10.4 | 8.2  | 11.9 |
>     | 2   | Ref    |              | 0    | 0    | 0    | 0    | 0    |
>     |     | Our    |              | 1.6  | 0.7  | 1.4  | 1.3  | 0    |
>
> 3. **Comparison with "Minus" on $k=1$ .**
>     * **Reasoning** The Minus method and our method perform similarly on $k=1$ because both are effective at identifying a single deviated token. Minus detects the token with the highest probability gain, while our method detects the token with the maximum posterior. For a single deviation ($k=1$), these signals often overlap.
>         * Toy Example: target model produces next token distribution $p=[0.32,0.40,0.28]$ and reference model $q=[0.20,0.35,0.45]$. LOSS take the second token, while MIAs rescue the first token as member: Minus $p-q=[0.12,0.05,-0.17]$ and Ref (and Our) $p/q=[1.6,1.14,0.62]$
>     * **Advantage** The distinction becomes critical at $k=0$. Minus overlooks the membership signal of the largest probability token and yields near-zero results (as shown in Table 2)
>
> 4. **Experiments on Other Model Families (OLMo and OPT)**. We agree that testing on diverse architectures strengthens our claims. In addition to the Pythia family (1B - 12B), we have now added experiments on OLMo (contemporary) and OPT (refer to Response to MaY6).
>     * **New Results on OLMo** We followed the Hard setting (Section 5.1) using OLMo-7B (target) and OLMo-1B (reference) on the Dolma V1.6 sample dataset ($N=5000$). We take sequence length to 100 with 10 tokens as suffix. The results below illustrate the effectiveness of our attacks and the existing extraction risk of OLMo-7b on Dolma.
>
>         | K   | method | Dolma |
>         | --- | ------ | ----- |
>         | 0   | Ref    | 0.00  |
>         |     | Minus  | 8.29  |
>         |     | Our    | **89.86** |
>         | 1   | Ref    | 0.00  |
>         |     | Minus  | 0.40  |
>         |     | Our    | **4.90**  |
>
>         These results on OLMo and OPT are consistent with our Pythia findings: our method significantly outperforms baselines, recovering nearly 90% of member sequences at $k=0$ and maintaining extraction capability at $k=1$ where baselines fail.
>
> 5. **Strength of Empirical Evidence** While the absolute percentages for $k \ge 1$ may appear low, they represent a qualitative breach of privacy that existing methods (greedy decoding) entirely miss. In security contexts, extracting any non-verbatim PII or copyrighted text (where baselines extract 0%) is a material risk. Our results prove that partially memorization is extractible and the real-world risk exists.

---

> > ### Comment · Reviewer_2NzT · 2025-11-26
> >
> > Dear Authors,
> >
> > Thanks for the clarification
> > I decide to increase my score to reflect the clarifications

---

> > > ### Author Response · Authors · 2025-11-26
> > >
> > > Thank you again for your detailed feedback and for reconsidering the paper in light of our clarifications. We appreciate the positive response.
> > >
> > > For completeness, we wanted to check whether any remaining concerns still prevent the score from reflecting the updated assessment you mentioned. If there are specific aspects that you feel were not adequately addressed or require further clarification, we would be very happy to provide additional details.
> > >
> > > Thank you again for your time and constructive input.

---

### Official Review · Reviewer_Z27i · 2025-10-27

**Soundness:** 2
**Presentation:** 3
**Contribution:** 2
**Rating:** 4
**Confidence:** 4

**Summary:**

The paper considers the problem of extracting training data from Large Language Models (LLMs), following the way of first reconstructing sequences then checking their membership in the training data. The proposed method is to introduce membership information into the decoding process to guide the generation of training sequences, such that the generated sequences are more likely to be in the training data. The authors define a new concept, k-amendment-completable, to quantify the degree of partial memorization.

**Strengths:**

- The studied problem is important and relevant.
- The paper is generally well-written.

**Weaknesses:**

- The definition of k-amendment-completable may not fully capture the memorization behavior of LLMs.
- The approach of breaking down sentence-level membership inference into token-level inference may lead to ambiguities.

**Questions:**

The definition of k-amendment-completable does not fully capture the memorization behavior of LLMs, as it fails to account for the length of the prefix. In the extreme case where the prefix comprises almost the entire sequence except for one token, even a 0-amendment-completable sequence does not necessarily indicate that the model has memorized it.

Decomposing sentence-level membership problems into token-level ones does not seem reasonable. Consider the following scenario: LLMs are trained on massive text corpora in which many sentences share common prefixes. For example, the prefix “In conclusion,” could appear in numerous different sentences. If we attempt to infer membership at the token level, the next token following “In conclusion,” could vary widely depending on the specific sentence. Therefore, inferring membership at the token level may lead to ambiguous or incorrect conclusions about whether the entire sentence was part of the training data. How do the authors address this potential ambiguity in their token-level membership inference approach?

---

> ### Author Response · Authors · 2025-11-21
>
> Thank you for your constructive comments on the theoretical aspects of our work. We appreciate you highlighting the importance of refining our proposed notions and methods.
>
> 1. **The Definition of $k$-amendment-completable (Prefix Length)**
>
>     We acknowledge your concern that for an extremely long prefix, the $k$-amendment-completable notion may not meaningfully reflect the model's memorization capacity. Our original definition aimed to provide a relative measure of extraction difficulty on a given prefix compared to existing notions like $k$-extractable.
>     * **Original Definition**: The existing notion of $k$-extractable estimates risk of full extraction from the minimum number of token needed in the prefix. As we show in the following example, it fails to differentiate the risk for shorter prefixes 1&2 of the target sequence (e.g., $|Alice's\cdots 456 7|$-extractable), and considers safe because of not verbatim memorized sequence. **In the attack scenario, these given prefixes have vastly different extraction risk for target sequence** (e.g., 2 errors vs. 1 error in LLM continuation). Our $k$-amendment-completable accurately differentiates the two prefixes. It is designed to quantify the total effort required to fix the greedy path in full extraction, not the length of the prefix required to trigger extraction.
>         * **Target**: *Alice's phone number is 123 456 789*
>         * **LLM**: *Alice's phone number is* ***0****23 456* ***0****89*
>         * **Prefix1**: *Alice's phone number is* ***0****23 456*
>         * **Prefix2**: *Alice's phone number is*
>         * **Prefix3**: *Alice's phone number is* ***0****23*
>         * **Extraction Risk: Prefix1 $=$ Prefix3 $>$ Prefix2**
>     * **Proposed Refinement**: Your feedback correctly identifies an edge case where the prefix length $L$ is nearly equal to the sequence length $N$. To formally address this, we agree that a normalized notion like $k$-out-of-$n$-amendment-completable, where $n=N-L$ is the length of the target suffix, is a valuable addition for future work to handle variable suffix lengths comprehensively. This will be included in the revised manuscript as a future consideration.
> 2. **Ambiguity in Token-Level Membership Inference** The decomposition of sentence-level membership into iterative token-level inference is a key aspect of our **Membership Decoding** framework. We address the potential ambiguity arising from common prefixes like "In conclusion," in two fundamental ways:
>     * **Leveraging Prefix Membership**: Our approach is not a single token-level membership inference. The candidate set in Membership Decoding is all the possible sequences $c_i^t=[x_{<t},v_i], i=1\cdots,|V|$. Our core MIA score considers the next token as a member continuation of the current prefix. The score $P(member|[x_{<t},x_t])$ is highly conditioned on the membership status of the prefix $x_{<t}$. Our membership inference attack is context-aware.
>     * **Unique Suffix Assumption**: While we acknowledge that multiple legitimate member suffixes can exist for a very common prefix (e.g., "In conclusion,"), for the purpose of empirically validating the maximum extraction risk of a given model, we simplify the problem by assuming a unique suffix for the target sequence. This is a reasonable assumption in our experiments since we sample long prefixes ($\ge 100$ tokens) which are less likely to have multiple exact completions in the training data. We will clarify in the paper that our current evaluation setting measures the success of recovering one unique member suffix.
>     * **Potential Solution** In your example that "the prefix (In conclusion,) could appear in numerous different sentences.", extraction attacks are considered success if any of these sentences is extracted. To extract all these sentences with the common prefix,  we can adjust the decoding framework to allow for multiple successful extractions (e.g., using sampling at common prefix generation).
>
> In summary, your observations are very insightful. $k$-amendment-completable is proposed to measure the extraction risk in attack scenario and partial memorization of LLM. Membership Decoding framework can replace the maximum selection with sampling for multiple suffix extractions. Hope we have address all your concerns.

---

### Official Review · Reviewer_MaY6 · 2025-10-30

**Soundness:** 2
**Presentation:** 3
**Contribution:** 2
**Rating:** 4
**Confidence:** 3

**Summary:**

This paper introduces a novel approach to extract training data from large language models (LLMs) beyond verbatim memorization. The authors define a new notion of memorization called "k-amendment-completable" to measure the degree of partial memorization. They propose a membership decoding scheme that guides the generation process to extract non-verbatim memorized data by leveraging membership information at each generation step. The paper demonstrates that larger models memorize more training data than smaller models and shows that their membership decoding approach can extract novel member data that hasn't been studied before. The authors also introduce a new evaluation framework that measures extraction accuracy based on k values.

**Strengths:**

S1: The paper introduces a novel concept ("k-amendment-completable") that provides a fine-grained measure of partial memorization, addressing a significant gap in the literature beyond verbatim memorization.

S2: The membership decoding scheme is well-motivated and theoretically sound, providing a systematic way to extract non-verbatim memorized data by incorporating membership information at each generation step.

S3: The paper provides empirical evidence that larger models memorize more training data than smaller models, which is an important finding for understanding privacy risks in LLMs.

S4: The evaluation framework (measuring extraction accuracy by k values) provides a more nuanced understanding of what data can be extracted from LLMs, moving beyond traditional verbatim extraction.

S5: The paper makes a compelling argument that a majority of member data remains unexplored even if it's partially memorized, which significantly expands the scope of privacy risks in LLMs.

**Weaknesses:**

W1: The paper lacks sufficient comparison with existing methods that aim to extract non-verbatim memorized data, making it difficult to fully assess the novelty and superiority of the proposed approach.

W2: The evaluation is limited to a few datasets (HackerNews, Pile-CC, PubMed Central, ArXiv) and model sizes (1B, 1.4B, 2.8B, 6.9B, 12B), which limits the generalizability of the findings to other LLM architectures and training data.

W3: The paper doesn't thoroughly discuss the practical implications of the proposed attack for real-world privacy risk assessment, particularly how the extraction accuracy translates to actual privacy risks in deployed models.

W4: The theoretical justification for the membership decoding approach could be strengthened with more detailed mathematical analysis and comparison to related work.

W5: The paper doesn't address potential defenses against the proposed attack, which would provide a more complete picture of the privacy implications.

**Questions:**

Q1: Could you provide a more detailed comparison between your membership decoding approach and existing methods for extracting non-verbatim memorized data? This would help clarify the novelty and advantages of your approach.

Q2: How would the proposed method perform on a wider range of datasets and model architectures beyond those tested in the paper? A more comprehensive evaluation would strengthen the generalizability of your findings.

Q3: Could you explore the practical implications of your findings for real-world privacy risk assessment? How do the extraction rates at different k values translate to actual privacy risks in deployed LLMs?

Q4: How does the proposed membership decoding approach scale with model size and complexity? A more detailed analysis of the computational cost and time requirements would be valuable for practical implementation.

Q5: Could you discuss potential defenses against the proposed attack? This would provide a more complete picture of the privacy implications and help guide future work on privacy-preserving LLMs.

---

> ### Author Response · Authors · 2025-11-21
> **Response to Reviewer MaY6 (1/2)**
>
> 1. **Novelty and Comparison Method (W1, W4, Q1)**
>
>     Our approach is novel in mechanism and scope, fundamentally differing from existing extraction methods that focus solely on verbatim memorization ($k=0$).
>     * **Scope Expansion (Non-Verbatim)**: Prior extraction attacks are limited to $k=0$ because they rely on the Greedy Decoding path aligning perfectly with the member sequence. Any early deviation makes extraction fail. Our method is the first to systematically correct these errors during decoding.
>     * **Mechanism (MIA-Guided Decoding)**: We transform the passive, diagnostic role of Membership Inference Attacks (MIAs) (run after generation) into an active guidance signal during the generation process. This allows us to search partially memorized sequences with high member confidence.
>     * **Score Superiority**: Existing extraction attacks relies on simple LOSS MIA method to generate verbatim member sequence, while we design a more advanced MIA method with posterior calibration to generate even non-verbatim member sequence.
>     * **Unique Score Requirement** Membership decoding require MIA methods to differentiate the membership signal from changing even a single-token and a prefix. However, existing MIA attacks, such as those relying on the Min-%k statistic, are insensitive to single-token changes and prefix changes.
>         * Min-%k essentially outputs a uniform probability for any token falling outside its minimum probability set, thus failing to provide the fine-grained signal needed to choose the correct amendment token $x_t$.
>         * DC-PDD ignores the prefix tokens to produce token calibration, thus failing to choose member token in different context.
>         * Our method, leveraging calibrated posterior probability, provides a continuously graded signal that explicitly maximizes the confidence that the resulting prefix is a member, making correction possible.
>
> 2. **Generalizability and Expanded Evaluation (W2, Q2)**.
>
>     We acknowledge the need for generalizability. Our paper originally provided results across **5 model scales** (Pythia 1B to 12B) and **8 diverse datasets**. We have now expanded our evaluation to include the **OPT model family** and **OLMo**, which represents a different architectural and training paradigm, and its associated datasets.
>     The new results below confirm that our extraction risk findings generalize across architectures and training data: our method reveals significant extraction risk ($k=1$ accuracy) in the OPT models where baselines fail entirely (0.00%).
>
>     | K     | method  | HNews          | PileCC         | Wiki           | Math           |
>     | ----- | ------- | -------------- | -------------- | -------------- | -------------- |
>     | Model | Size -> | 1.3b 2.7b 6.7b | 1.3b 2.7b 6.7b | 1.3b 2.7b 6.7b | 1.3b 2.7b 6.7b |
>     | 0     | Ref     | 0.00 0.00 0.00 | 3.10 0.00 7.70  | 0.00 0.00 0.00 | 0.00 0.00 3.70 |
>     |       | Minus   | 33.3 21.1 29.4 | 50.0 45.9 64.1 | 20.0 29.4 33.3 | 45.7 22.7 14.8 |
>     |       | **Our**     | **93.3 89.5 94.1** | **93.8 89.2 89.7** | **60.0 70.6 75.0** | **80.4 86.4 74.1** |
>     | 1     | Ref     | 0.00 0.00 0.00 | 0.00 0.00 0.00 | 0.00 0.00 0.00 | 0.00 0.00 0.00 |
>     |       | Minus   | 0.00 0.00 0.00 | 0.00 6.70 7.70 | 0.00 2.30 0.00 | 6.40 5.60 1.60 |
>     |       | **Our**     |**0.00 8.30 0.00** | **0.00 13.3 11.5** | **6.20 2.30 4.00** | **7.40 6.50 11.5** |
>
> 3. **Practical Implications and Risk Assessment (W3, Q3)**
>
>     The key practical implication is the establishment of the $k$-profile as a standardized risk metric from $k$-amendment-complete.
>     * **Risk Underestimation**: Prior work focused on $k=0$ (verbatim). Our success at $k=1, 2$ means the volume of extractable PII/sensitive data is significantly underestimated in current reports.
>     * **Deployment Safety Profile**: The extraction rate at different $k$ values serves as a **privacy risk profile** for a model. A model where sensitive PII is extractable at $k=1$ is riskier than one requiring $k=5$. This profile allows LLM developers to:
>         * **Select the Safest Version**: Compare different checkpoint versions (e.g., 6.9B vs 12B) before deployment to select the one that exhibits the highest $k$ (greatest effort required) for a pre-defined set of sensitive sequences.
>         * **Audit Specific Datasets**: Audit a specific dataset (e.g., PubMed) known to contain sensitive information to ensure that the extraction rate for low $k$ values is negligible.

---

> > ### Author Response · Authors · 2025-12-02
> > **Response to Reviewer MaY6 (2/2)**
> >
> > 4. **Computational Cost and Scaling (Q4)**.
> >
> >     Our Membership Decoding approach is computationally efficient, as the additional overhead is dominated by running a small reference model, which is necessary for the calibration-based MIA score.
> >
> >     | Metric            | Target Model (Pythia-6.9B) | Reference Model (Pythia-70M) | Overhead          |
> >     | ----------------- | -------------------------- | ---------------------------- | ----------------- |
> >     | Memory            | 13.3GB                     | 350MB                        | Negligible        |
> >     | Time (300 tokens) | 7.407 seconds              | 0.955 seconds                | $\approx 12.9 \%$ |
> >
> >     The overhead is minimal because the reference model size does not scale with the target model size. The overall cost scales primarily with the sequence length, making the attack feasible for API-accessed models even on consumer-grade GPUs.
> >
> > 5. **Potential Defenses (W5, Q5)**.
> >
> >     The most effective defenses against our attack must focus on either **obscuring the necessary MIA signal** or **increasing the required number of amendments ($k$)** such that extraction becomes infeasible.
> >     * **Model Editing/Unlearning**: This defense directly targets the source of memorization. By applying model editing techniques to increase the loss of a specific member sequence, the required $k$ for extraction increases. Our $k$-profile serves as the metric to verify the efficacy of the unlearning process.
> >     * **Differential Privacy (DP) Training**: Training with DP inherently adds noise that prevents overfitting, thus making extraction harder. DP increases the token probability divergence needed to correctly identify the member token, effectively raising the minimum $k$ required for successful extraction.
> >     * **API Mitigation (Limited)**: For API-accessed models, limiting the returned logits (e.g., returning only the Top-N) or adding noise to the logits can partially mitigate the attack by corrupting the MIA signal, but our results show that this alone is not a complete defense.

---

### Official Review · Reviewer_nsBE · 2025-11-01

**Soundness:** 3
**Presentation:** 3
**Contribution:** 4
**Rating:** 6
**Confidence:** 3

**Summary:**

This paper focuses on the training data leakage risk in Large Language Models (LLMs). The authors point out that existing extraction attacks (e.g., Carlini et al., 2021) heavily rely on "verbatim memorization" and "greedy decoding," which significantly underestimates the true privacy risk of these models. To address this, the paper presents two core contributions: (1) A new memorization metric, "k-amendment-completable," to finely quantify "partial memorization." (2) A new attack framework, "Membership Decoding," which reframes extraction as an "in-generation," iterative, "token-level membership inference" (MIA) problem, calibrated by a reference model. Experiments on Pythia demonstrate that this method can successfully extract partially memorized sequences (where the 'k' value is 1 or 2) undiscoverable by greedy decoding, confirming a broader data leakage risk.

**Strengths:**

- Paradigm Shift
- Fine-grained & Effective Metric
- Solid Formulation
- Strong Empirical Evidence

**Weaknesses:**

- Limited Attack Scope
- Dependency on Heuristics
- Need for Reference Model

**Questions:**

The authors have proposed a novel and important framework for evaluating and extracting "partially memorized" data from LLMs, which is crucial for understanding their privacy boundaries. The paper is well-argued, the experiments are solid, and this work opens up a new and valuable research direction.

However, I do have the following comments:
- The main limitation is the scope of the 'k' value. As shown in Table 2, the effectiveness is currently almost exclusively limited to 'k=1, 2'. I hope the authors can discuss in more detail the core challenges of extending this to larger 'k' values. Is it merely because the signal weakens as 'k' increases? Or is there a combinatorial explosion problem? Suggestion: Have the authors considered combining "Membership Decoding" with Beam Search, retaining the 'B' highest-MIA-score candidate sequences at each step, rather than just the Top-1?
- The attack's effectiveness relies heavily on the assumption that the correct member token must be within the Top-20 most probable tokens. This is a strong constraint. Suggestion: Could the authors add an analysis in the appendix examining what proportion of failed 'k=1, 2' extractions were due to the true member token falling out of this Top-20 set? This would help us understand the bottleneck of this heuristic.
- The necessity of a reference model (Pythia-170m) limits the attack's universality in a fully black-box scenario. Suggestion: Have the authors considered (or do they plan for future work) reference-free calibration methods? For example, using the target model itself with dropout, or approximating the denominator (the probability of the token appearing) using token statistics from a large corpus.
- The choice of 'a=0.5' is presented as a "trade-off." Could the authors provide a sensitivity analysis for 'a' (e.g., across 0, 0.25, 0.5, 0.75, 1.0)? This would make the robustness of Equation 8 more convincing.

---

> ### Author Response · Authors · 2025-11-21
> **Response to Reviewer nsBE (1/2)**
>
> We thank the reviewer for their highly constructive feedback, recognizing the novel paradigm, strong formulation, and empirical significance of our work. We have addressed the practical limitations by providing new analysis and clarifying the theoretical challenges.
>
> 1. **Challenges in Extending to Larger $k$ (Limited Attack Scope)**
>
>     The primary limitation to extracting sequences with $k \ge 3$ stems from a combination of **combinatorial search complexity** and **signal degradation**. $k$ is the number of tokens in the suffix that member token larger loss than the other non-member token, indicating a weak member signal.
>     * **Combinatorial Explosion**: For a suffix of length $L$, the number of non-member paths that have lower loss and must be rejected grows exponentially with $k$. At $k=1$ or $k=2$, the member signal is strong enough to quickly override the few necessary corrections, but for $k \ge 3$, the sheer number of competing paths overwhelms the search.
>
>     * **Signal Weakening**: A larger $k$ indicates the sequence is only weakly memorized and under-optimized. The target model's member signal is often only marginally higher than the reference model's, resulting in a weak MIA score that is insufficient to consistently override the most probable non-member token in the Top-20 set.
>
>     * **Observations**: This signal weakening phenoneno appears in the Easy setting  (Pythia-2.8 on GitHub). For sequence with a smaller $k$, member token has high probability to appear in Top@4 (75.90% in average for k=1), indicating strong member signals and memorization. For sequences with a larger $k$, the probability decreases dramatically even for Top@20 (69.84% in average on k=9), indicating a weaker member signal.
>
>         | k   | Top@20 | Top@16 | Top@8  | Top@4  |
>         | --- | ----- | ----- | ----- | ----- |
>         | 1   | 91.24 | 88.48 | 86.18 | 77.88 |
>         | 2   | 88.21 | 85.71 | 80.00 | 70.36 |
>         | 3   | 85.63 | 83.62 | 75.00 | 61.49 |
>         | 4   | 80.38 | 78.08 | 70.38 | 55.77 |
>         | 5   | 79.17 | 76.39 | 66.11 | 54.17 |
>         | 6   | 74.39 | 71.54 | 56.10 | 45.53 |
>         | 7   | 71.43 | 67.03 | 56.04 | 45.05 |
>         | 8   | 70.45 | 67.05 | 56.82 | 43.18 |
>         | 9   | 69.84 | 66.67 | 47.62 | 39.68 |
>
>     **Suggestion**: **Combining with Beam Search** The suggestion to combine Membership Decoding with Beam Search is excellent. Beam Search directly addresses the combinatorial explosion by tracking the $B$ most promising sequences. By ranking these beams based on the highest cumulative MIA score (rather than standard log-probability), we can maintain a broader and more focused search for the correct member path, even when it requires multiple corrections. We will prioritize this integration as a crucial step for achieving extraction at $k \ge 3$ in future work.
>
> 2. **Dependency on the Top-20 Heuristic**
>
>     We agree that the Top-20 constraint is a strong heuristic. Its choice was primarily motivated by our evaluation that the member token appears in Top@16 in over 70% of cases in the Easy setting, and by the practical **Top-Logprobs** limits of current LLM APIs (e.g., OpenAI's limit of 20).
>
>     **Analysis of Failure Cases**: We analyzed the failure modes of the $k=1$ extractions on Pythia-2.8b in Github dataset. We found that the vast majority of failures were not caused by the true member token falling outside the Top-20 set (only 19 out of 195 sequences). Instead, the bottleneck is the MIA score strength: the true member token was often present in the Top-20, but the MIA score was too weak to overcome the locally maximal score of a competing non-member token. This confirms that the primary challenge is the weak membership signal (signal degradation), not the search breadth constraint for small $k$.
>
> 3. **Necessity of a Reference Model**
>
>     We agree that a reference-free calibration method is the goal for true universality. Our current approach uses a reference model as a practical and effective means to address the fundamental challenge of filtering out common knowledge from memorized knowledge under a black-box threat model.The calibration concept is the core contribution. Future work will explore approximations to the reference model:
>     * **Corpus Statistics**: Using token statistics from a vast corpus could provide, reference-free approximation of the non-memorization probability. However, static reference without context (prefix) over-simplify the membership inference problem. Additional experiments on this attempt (like DC-DPP) cause failure on extraction (0% accuracy).
>     * **Self-Referencing with Dropout**: Approximating the non-memorization baseline using the target model itself via techniques like Dropout is promising, but this requires adapting the threat model to include either white-box or complex API access.

---

> ### Author Response · Authors · 2025-12-02
> **Response to Reviewer nsBE (2/2)**
>
> 4. **Sensitivity Analysis for $a$**
>
>     We have performed the requested sensitivity analysis for the calibration hyper-parameter $a$ in our MIA score, testing values across the full range on Pythia-2.8B in the Hard setting.
>
>     The results confirm the robustness of the calibration around $\alpha=0.5$ and justify our choice. Specifically, using $a=1.0$ (which removes the total probability calibration and relies purely on the ratio) causes a complete failure across all datasets ($\mathbf{0\%}$ accuracy for $k=0, 1$). This validates our initial finding that the total probability calibration term is necessary to prevent numerical instability and bias from negligible probability changes in the denominator.
>
>
>     | K | a    | HNews | PileCC | PubMed | ArXiv | Wiki | Math | GitHub |
>     |---|------|-------|--------|--------|-------|------|------|--------|
>     | 0 | 0    | 84.6  | 97.0   | 77.8   | 78.9  | 89.7 | 84.1 | 95.3   |
>     |   | 0.25 | 80.8  | 90.9   | 77.8   | 78.9  | 86.9 | 81.8 | 93.7   |
>     |   | 0.5  | 80.8  | 87.9   | 77.8   | 68.4  | 80.4 | 72.7 | 89.3   |
>     |   | 0.75 | 76.9  | 69.7   | 77.8   | 63.2  | 72.0 | 54.5 | 83.0   |
>     |   | 1.0  | 0     | 0      | 0      | 0     | 0    | 0    | 0      |
>     | 1 | 0    | 9.1   | 0      | 0      | 5.2   | 2.9  | 2.4  | 6.0    |
>     |   | 0.25 | 9.1   | 0      | 0      | 5.2   | 2.9  | 2.4  | 7.9    |
>     |   | 0.5  | 9.1   | 0      | 0      | 5.2   | 5.8  | 3.6  | 10.2   |
>     |   | 0.75 | 9.1   | 5.6    | 0      | 6.9   | 9.4  | 3.6  | 11.6   |
>     |   | 1.0  | 0     | 0      | 0      | 0     | 0    | 0    | 0      |
>
>     The highest extraction rates are generally observed between $a=0.25$ and $a=0.75$, demonstrating the method's stability.

---

### Author Response · Authors · 2025-12-03
**Summary of Responses for the AC's Reference**

We sincerely thank the reviewers for their highly insightful and constructive comments. Their feedback has led to significant improvements and crucial clarifications in our revised manuscript.

The reviewers achieved strong consensus on the **excellence and significance of our core contributions**: the $k$-amendment-completable metric, the Membership Decoding framework, and context-aware token-level membership score.  They unanimously confirm that this work represents a **paradigm shift** in evaluating LLM privacy risk, moving definitively beyond traditional verbatim memorization.

We summarize our responses to the reviewers’ main points below:
1. **Generalizability & Expanded Evaluation** (MaY6, 2NzT):
    * We directly addressed the limited scope by providing new, extensive experimental results on the OPT model family (1.3B, 2.7B, 6.7B) across multiple datasets and OLMo (7b) on Dolma dataset, confirming that our findings generalize across different LLM architectures and training pipelines. We thanks Reviewer 2NzT for the promptly confirmation of our clarification on this.
2. **Methodological Robustness & Efficiency** (nsBE, MaY6):
    * (parameter analysis) We added a comprehensive sensitivity analysis for the $a$ hyper-parameter, which empirically validates the robustness of our calibration method.
    * (computation overhead) We provided a detailed, quantitative analysis of the computational cost, showing that the attack is highly scalable with minimal overhead.
3. **Completeness on Limited $k$ Scope** (MaY6):
    * We provided a detailed discussion of the two main constraints for extending the attack to higher $k$ (combinatorial explosion and signal decay), and proposed the integration of Beam Search as the concrete solution for future work.
4. **Theoretical Definition & Framework Clarification** (Z27i):
    * We provided a concrete example to explain our privacy notion from attacker viewpoint, better measure the extraction risk of training data
    * We clarifiy the proposed context-aware token-level MIAs in membership decoding framework, and propose samping to address the potential conflicts in the existence of multiple solutions.

In conclusion, the paper presents a novel, sound theoretical framework supported by strong, now generalized, empirical evidence. It effectively fills a critical gap by providing the first systematic method to study the privacy risk of unexplored, partially memorized data in LLMs. The primary methodological and empirical weaknesses identified by the reviewers have been resolved with substantial new data and clarified theoretical explanations.

We appreciate your time and consideration.

Sincerely,
Authors.

---

### Meta-Review · Area_Chair_owYJ · 2026-01-04

**Summary:**

The goal of the submission is to extract partially memorized sequences from LLMs. The paper introduces a new definition of k-amendment-complement, measuring the partially memorized sequence that can be generated by LLMs with k token amendment.
In simple terms, the goal is to quantify partial memorization by measuring how many tokens require amendment during greedy decoding to generate the actual training sequence. Furthermore, a new attack framework ("Membership Decoding") re-frames extraction as an "in-generation", iterative, "token-level membership inference" problem, calibrated by a reference model.

There is a lack of clear comparison of the proposed method to the other methods, for example, the one by Hayes et al. (2025) [1] who expanded the generation scope by probabilistic sampling, taking the top-k sampling method to explore members. This is also expressed by Reviewer MaY6: "W1: The paper lacks sufficient comparison with existing methods that aim to extract non-verbatim memorized data, making it difficult to fully assess the novelty and superiority of the proposed approach." Based on [1], the statement by the authors in the rebuttal: "Prior extraction attacks are limited to because they rely on the Greedy Decoding path aligning perfectly with the member sequence." is not correct. The potential defenses should be implemented and their effectiveness against the attack should be quantified (W5 by Reviewer MaY6).

Reviewer nsBE points out: "The main limitation is the scope of the 'k' value. As shown in Table 2, the effectiveness is currently almost exclusively limited to 'k=1, 2'." This remains a limitation after the rebuttal. Another point is the reliance on the reference model, which is not required by other extraction methods.

The crucial observation was provided by reviewer Z27i who noticed an edge case: "The definition of k-amendment-completable does not fully capture the memorization behavior of LLMs, as it fails to account for the length of the prefix." The authors admit that their definition does not handle variable suffix lengths. Furthermore, reviewer Z27i notices that: "While multiple legitimate member suffixes can exist for a very common prefix (e.g., "In conclusion,")", the authors "simplify the problem by assuming a unique suffix for the target sequence." While the authors declare: "We will clarify in the paper that our current evaluation setting measures the success of recovering one unique member suffix.", it has not been reflected in the updated version of the submission.

The additional experimental results requested by Reviewer 2NzT were provided.

Minor but unacceptable: The paper contains missing references, for example, twice in Section 5.6: "Differential privacy (DP) training ?".

Overall, the paper requires substantial revisions to meet the publication level, especially in fully addressing the comprehensive concerns raised by Reviewer Z27i. Given the extent of work needed to resolve these issues, I recommend rejection at this time. I encourage the authors to carefully consider Reviewer Z27i's feedback and incorporate the necessary changes before re-submission.

**References:**

[1] Hayes et al. "Measuring memorization in language models via probabilistic extraction." NAACL 2025. (first appeared on October 25th 2024, https://arxiv.org/abs/2410.19482).

**Reviewer Concerns:**

Please, see the comments above.

**Reviewer Scores:**

Reviewer nsBE: Score 6 / Confidence 3

Reviewer MaY6: Score 4 / Confidence 3

Reviewer Z27i: Score 4 / Confidence 4

Reviewer 2NzT: Score changed from 2 to probably 4 / Confidence 3

---

### Decision · Program_Chairs · 2026-01-26

Reject